# Modified Two-Point Correction Method for Wide-Spectrum LWIR Detection System

**DOI:** 10.3390/s23042054

**Published:** 2023-02-11

**Authors:** Di Zhang, He Sun, Dejiang Wang, Jinghong Liu, Cheng Chen

**Affiliations:** 1Changchun Institute of Optics, Fine Mechanics and Physics, Chinese Academy of Sciences, Changchun 130033, China; 2University of Chinese Academy of Sciences, Beijing 100049, China

**Keywords:** atmospheric transmittance, long wave infrared, non-uniformity correction, two-point correction, fixed-pattern noise

## Abstract

Non-uniformity commonly exists in the infrared focal plane, which behaves as the fixed-pattern noise (FPN) and seriously affects the image quality of long-wave infrared (LWIR) detection systems. The two-point correction (TPC) method is commonly used to reduce image FPN in engineering. However, when a wide-spectrum LWIR detection system calibrated with a black body is used to detect weak and small targets in the sky, FPN still appears in the image, affecting its uniformity. The effects of atmospheric transmittance characteristics of long-range paths on the non-uniformity of wide-spectrum long-wave infrared systems have not been studied. This paper proposes a modified TPC model based on spectral subdivision that introduces atmospheric transmittance. Additionally, the effects of atmospheric transmittance characteristics on the long-wave infrared non-uniform correction coefficient are analyzed. The experimental results for a black body scene and sky scene using a weak and small target detection system with a long-wave Sofradir FPA demonstrate that the wide-spectrum LWIR detection system fully considers atmospheric transmittance when performing calibration based on the TPC method, which can reduce the non-uniformity of the image.

## 1. Introduction

Infrared focal-plane arrays (FPAs) are widely used in airborne infrared search and track (IRST) systems [1,2,3], which possess night vision, anti-hidden, and mist-penetrating capabilities. However, under the same infrared irradiance, the responses between the different units within an IRFPA can vary due to the external environment, infrared sensitive components, circuit structures and semiconductor characteristics. This manifests as fixed-pattern noise (FPN) in the infrared image, which is the source of non-uniformity in IRFPA [4]. Non-uniform noise seriously degrades the imaging quality of the system, reduces the system resolution and point target signal-to-noise ratio (SNR), which is the bottleneck preventing the infrared point target detection system reaching the background limit [5,6]. Therefore, it is necessary to perform non-uniformity correction (NUC) on the acquired infrared image for subsequent successful detection of weak and small targets.

NUC methods can be broadly classified into two major categories. The first category includes the most commonly used blackbody-based NUC (BBNUC) methods, where all pixels are linearly mapped to the average response of a uniform radiance source [7]. The second category comprises scene-based NUC (SBNUC) methods, which are statistics-based methods, registration based methods, temporal filtering-based methods and optimal estimation-based methods [8]. Statistical methods rely on spatio-temporal assumptions but are motion dependent. Registration-based methods assume that different pixels respond identically to the same scene point within certain blocks of time [9,10,11]. However, accurate registration in IRST is difficult due to the low contrast of scenes filled with the sky, as well as image motion caused by scanning and high levels of vibration. In the practical application of airborne IRST, because the platform itself moves at high speed, and the enemy targets (fighters, missiles, etc.) to be captured are usually also extremely maneuverable, the relative motion between the target and the imaging sensor changes abruptly, making it difficult to register individual frames in the sequence image, especially in the working state of quick search. The temporal high-pass filtering (THP) based method [12] uses high-pass filtering in the temporal domain to remove the FPN based on its low frequency characteristics. However, static object features will also be removed when using the THP method, which results in serious ghosting artifacts. Although, in seeking to overcome this problem, some improved methods such as the space low-pass and temporal high-pass (SLTH) algorithm [13] and the bilateral-filter-based temporal high-pass (BFTH) algorithm [14] have been developed, the cut-off frequency of the spatial filter and the temporal filter are difficult to determine, and ghosting artifacts still exist. SBNUC methods offer better correction effects than the BBNUC method for some scenes. However, these methods are complex, require a large amount of calculations and storage space, and are not as robust as the two-point correction method. In short, the scene method is overly reliant on the scene and target imaged by the system. Moreover, it is often necessary to classify the target and scene and use different residual scene elimination algorithms, making this method difficult to use in unattended automatic infrared target recognition and detection systems.

Linear two-point calibration (TPC). TPC algorithm is a commonly used BBNUC method. The TPC algorithm is well known and allows to compensate for both gain and offset variations of particular pixels in the array. Moreover, TPC is easy to implement by hardware and quite sufficient in many applications. In the detection of weak and small targets in the long-distance sky background, it is necessary to select two reference temperature points within the temperature range corresponding to the gradient scene for calibration, in order to reduce the non-linearity caused by the FPA and optical system. Therefore, the accuracy of the correction coefficients is key to minimizing fixed-pattern noise.

In practice, we found that correcting the sky image with the BBNUC coefficients still results in fixed-pattern noise. Because the ambient temperature of the system remains unchanged, we suspect that the radiance of the sky scene is inconsistent with the blackbody radiance resulting in the appearance of fixed-pattern noise. However, the impact of atmospheric transmittance on two-point correction method for wide-spectrum LWIR detection system has never been reported.

Our research aims to analyze the influence of atmospheric transmittance on TPC for a wide-spectrum LWIR detection system, and compensate both gain and offset through BBNUC method to compensate the corresponding detector difference caused by spectral differences. In Section 2, the relationship between detector responsivity and the two-point correction coefficient is derived, but there are many factors affecting detector responsivity. Mathematical modeling is required at the photonic level and spectral segmentation level to more deeply analyze the variables affecting the two-point correction coefficient. In Section 3, a modified NUC model based on spectral subdivision is proposed based on detector spectral response and target spectral radiation. The relationship between the correction coefficient of the two-point method and atmospheric transmittance is analyzed in a physical sense. For Section 4, we customized a filter to simulate the atmospheric transmittance of an actual experimental environment according to the response wavelength of 8–12 μm. A comparative experiment with and without the simulated atmospheric transmittance filter was then carried out. The influence of spectrum on detector nonuniformity correction coefficient was verified by comparing the standard deviation of the two groups of correction coefficients in a comparative experiment. The experimental results show that when viewing the sky scene with the calibrated coefficient after adding the filter, the fixed-pattern noise is significantly weakened. We then performed multiscale analyses of local region standard deviation, which proves that when the wide-spectrum LWIR detection system is calibrated via the modified two-point correction method, the non-uniformity of the image is reduced, and the image quality can be improved in actual aerial imaging only by fully considering the characteristics of the atmospheric transmittance.

## 2. Traditional Two-Point Calibration Model

The most commonly used response model of the IRFPA is:(1)$$I_{i,j}=G_{i,j}\Phi_{i,j}+O_{i,j}$$
where *I_i,j_* is the digital gray value output by the detector, *G_i,_*_j_ is the response gain of each pixel, *O_i,_*_j_ is the response offset of each pixel, and *Φ_i,j_* is the radiation flux received by the IRFPA pixel. The IRFPA responses are non-uniform, since each pixel *G_i,j_* and *O_i,j_* are different.

In the TPC method, the black body temperature is set as *T*_1_ and *T*_2_, corresponding to the two obtained images *I*_1_ and *I*_2_, and the radiant flux incident to each pixel is *Φ*_1_ and *Φ*_2_ respectively, then,
(2)$$\{\begin{array}{c} I_{1,i,j}=G_{i,j}\Phi_{1}+O_{i,j}\\ I_{2,i,j}=G_{i,j}\Phi_{2}+O_{i,j} \end{array}$$

Assuming that the correction gain coefficient is *K_i,j_* and the correction bias coefficient *B_i,j_*, then,
(3)$$\{\begin{array}{c} \bar{I_{1}}=K_{i,j}\Phi_{1}+B_{i,j}\\ \bar{I_{2}}=K_{i,j}\Phi_{2}+B_{i,j} \end{array}$$
where $\bar{I_{1}}$ and $\bar{I_{2}}$ are the image mean value. Next, the TPC gain coefficient *K_i,j_* and offset coefficient *B_i,j_* can be calculated as:(4)$$\{\begin{array}{c} K_{i,j}=\frac{\bar{I_{1}}-\bar{I_{2}}}{I_{1,i,j}-I_{2,i,j}}\\ B_{i,j}=\frac{I_{1,i,j}\bar{I_{2}}-I_{2,i,j}\bar{I_{1}}}{I_{1,i,j}-I_{2,i,j}} \end{array}$$

By inserting Equations (2) and (3) into Equation (4), the physical expression of the correction coefficient of the two-point method can be expressed as:(5)$$\{\begin{array}{c} K_{i,j}=\frac{\bar{G}}{G_{i,j}}\\ B_{i,j}=\bar{O}-\frac{\bar{G}}{G_{i,j}}O_{i,j} \end{array}$$

From the physical expression of the correction coefficient of the traditional TPC model, it can be seen that when considering the factors affecting the calibration parameters, only the average response parameter of blackbody radiation flux received by the IRFPA is usually considered, while the influence of atmospheric transmittance on the radiation flux received by the IRFPA is ignored.

## 3. Modified TPC Method Based on Spectral Subdivision

In this section, a modified model of infrared detector response is established by introducing atmospheric transmittance into the traditional response model. Then, the modified model and the traditional model are simulated and analyzed. The influence of atmospheric transmittance on the accuracy of correction coefficient is verified mathematically.

### 3.1. Modified Model

The IRFPAs commonly used in the imaging and detection of military targets in the field are photon detectors. The electrons in the detector material directly absorb the energy of the incident infrared radiation photons, such that the motion states of the electrons change, thereby realizing photoelectric conversion. Due to the electron absorption transition, such detectors are wavelength-selective and have a faster response and higher sensitivity than thermal detectors [15,16]. Atmospheric radiation enters the IRFPA after passing through the optical system. The atmospheric radiation flux received by a single pixel can be expressed as *Φ*(*λ*), and within the integration time *t*, the number of photons incident to a single pixel *n_p_* can be expressed as:(6)$$n_{p}=\frac{\Phi \left(\lambda \right)\cdot t\cdot \lambda }{h\cdot c}$$
where *h* is Planck’s constant and *c* is the speed of light. The photons incident to the IRFPA are absorbed and converted into charges by the photosensitive element, and the quantum conversion efficiency is *η*(*λ*). Therefore, the number of charges *n_e_* accumulated by photon radiation can be expressed as:(7)$$n_{e}=\eta (\lambda )\cdot n_{p}$$

After the accumulated charge is quantized by the readout circuit, the digital gray value *X* is output as:(8)$$\begin{array}{cc} X & =kn_{e}+b\\ & =\frac{k\cdot \eta (\lambda )\cdot \Phi \left(\lambda \right)\cdot t\cdot \lambda }{h\cdot c}+b \end{array}$$
where *k* is the global linear gain of the detector, and *b* is the pixel offset, in the actual optical imaging system, the response band of the detector will not be a certain wavelength but has a certain band range. Therefore, it is necessary to integrate the above-mentioned wavelength-related quantities into the range of the response band *λ1*~*λ2*. Additionally, there are differences in the spectral response coefficients of different pixels. Then the output gray value *X*(*λ*) of the detector can be expressed as:(9)$$X\left(\lambda \right)\;=\int_{\lambda_{1}}^{\lambda_{2}}\frac{k(\lambda )\cdot \eta (\lambda )\cdot \Phi \left(\lambda \right)\cdot t\cdot }{h\cdot c}d\lambda +b$$

The black body used in the TPC method can be equivalent to a gray body, and its emissivity is a uniform curve in each frequency band, while the atmospheric radiation is a selective radiator whose emissivity is related to wavelength and temperature. The distribution of spectral radiation indicates there is selective radiation at different wavelengths, and it is related to the wavelength. The distribution curves of the spectral emissivity and spectral radiation output of black body, gray body, and selective radiator are shown in Figure 1.

In the case of the same gray value of the detector response, there is a large difference in the radiative output of the sky background and the black body in the response frequency band. Therefore, the influence of atmospheric transmittance *τ_s_*(*λ*) on the gray value of the image should be considered when calibrating the infrared detection system. By compensating for the atmospheric transmittance in the response model of the IRFPA, we obtain.
(10)$$X(\lambda )=\int_{\lambda_{1}}^{\lambda_{2}}\frac{k(\lambda )\cdot \tau_{S}\left(\lambda \right)\cdot \eta (\lambda )\cdot \Phi \left(\lambda \right)\cdot t\cdot }{h\cdot c}d\lambda +b$$

For the pixel (*i*, *j*) of the IRFPA, Equation (10), is equivalent to:(11)$$X_{i,j}(\lambda )=\int_{\lambda_{1}}^{\lambda_{2}}k_{i,j}\left(\lambda \right)\tau_{S}\left(\lambda \right)\eta \left(\lambda \right)\Phi \left(\lambda \right)d\lambda +b_{i,j}$$
where $k_{i,j}=\frac{kt}{hc}$. The two-point calibration method uses the pixel output mean value $\bar{X_{i,j}^{L}(\lambda )}$ of the low temperature black body and the pixel output mean value $\bar{X_{i,j}^{H}(\lambda )}$ of the high temperature black body to correct the gain coefficient and offset coefficient, so that all pixels in the output between the high and low temperature points form a straight line. The specific expression of the two-point correction method is as follows:(12)$$\{\begin{array}{c} \bar{X_{{}_{i,j}}^{L}(\lambda )}=K_{i,j}X_{{}_{i,j}}^{L}(\lambda )+B_{i,j}\\ \bar{X_{{}_{i,j}}^{H}(\lambda )}=K_{i,j}X_{{}_{i,j}}^{H}(\lambda )+B_{i,j} \end{array}$$

The physical expressions of the correction gain coefficient and correction offset coefficient are obtained as:(13)$$\{\begin{array}{c} K_{i,j}=\frac{\left[\sum_{i=1}^{M}\sum_{j=1}^{N}\int_{\lambda_{1}}^{\lambda_{2}}k_{i,j}(\lambda )\tau_{S}\left(\lambda \right)\eta \left(\lambda \right)\Phi_{i,j}^{H}\left(\lambda \right)d\lambda -\sum_{i=1}^{M}\sum_{j=1}^{N}\int_{\lambda_{1}}^{\lambda_{2}}k_{i,j}(\lambda )\tau_{S}\left(\lambda \right)\eta \left(\lambda \right)\Phi_{i,j}^{L}\left(\lambda \right)d\lambda \right]}{M\times N\left(\int_{\lambda_{1}}^{\lambda_{2}}k_{i,j}(\lambda )\tau_{S}\left(\lambda \right)\eta \left(\lambda \right)\Phi_{i,j}^{H}\left(\lambda \right)d\lambda -\int_{\lambda_{1}}^{\lambda_{2}}k_{i,j}(\lambda )\tau_{S}\left(\lambda \right)\eta \left(\lambda \right)\Phi_{i,j}^{L}\left(\lambda \right)d\lambda \right)}\\ B_{i,j}=\bar{b_{i,j}}+\frac{1}{MN}\sum_{i=1}^{M}\sum_{j=1}^{N}\int_{\lambda_{1}}^{\lambda_{2}}k_{i,j}(\lambda )\tau_{S}\left(\lambda \right)\eta \left(\lambda \right)\Phi_{i,j}^{L}\left(\lambda \right)d\lambda -K_{i,j}\left[\int_{\lambda_{1}}^{\lambda_{2}}k_{i,j}(\lambda )\tau_{S}\left(\lambda \right)\eta \left(\lambda \right)\Phi_{i,j}^{L}\left(\lambda \right)d\lambda +b_{i,j}\right] \end{array}$$
where *M* × *N* is the resolution of IRFPA, $\Phi_{i,j}^{L}\left(\lambda \right)$ is the radiant flux of the low temperature scenario, and $\Phi_{i,j}^{H}\left(\lambda \right)$ is the radiant flux of the high temperature scenario. Based on Equation (13), the TPC method correction coefficients are related to the atmospheric transmittance, while the atmospheric transmittance will vary greatly in different working scenarios. As a result, the atmospheric radiation flux received by the IRFPA is quite different from the black body radiation flux. Taking the response wavelength-band of the 8~12 μm long-wave IRFPA as an example, we simulated the radiation flux received by the IRFPA in different working scenario, as shown in Figure 2.

The above theoretical analysis shows that the radiations flux received by the IRFPA varies greatly due to the influence of atmospheric transmittance in different scenarios. For the black body TPC method, atmospheric transmittance in the response frequency band of the IRFPA can be approximated to 1. When the observation background of the infrared detection system is the sky scene, the attenuation of infrared radiation depends on the radiation wavelength, path length, content of atmospheric components, and atmospheric environment [17]. The influence of the atmospheric transmittance of the infrared radiation path cannot be ignored. Infrared correction parameters will be affected by atmospheric transmittance. Therefore, when using the infrared focal plane array calibrated by the black body to detect the target in the sky background, there will be fixed-pattern noise caused by inconsistency between atmospheric radiation and black body radiation.

### 3.2. Performance Analysis

The difference between the modified two-point correction model and the traditional two-point correction model is mainly due to the consideration of the influence of atmospheric transmittance on the spectral response parameters of the infrared detector. In order to analyze the influence of the modified model on the correction coefficients more scientifically, we compare and simulate the correction coefficient of the traditional infrared correction model and the modified model.

Since long-wave infrared detector generally has a high sensitivity in the central region, we adopt the Gaussian distribution to simulate the global gain *k_i,j_* of the detector,
(14)$$k(x,y)=\frac{1}{2\pi \sigma^{2}}exp\left[-\frac{1}{2}\left(\frac{{\left(x-\mu_{1}\right)}^{2}+{\left(x-\mu_{2}\right)}^{2}}{\sigma^{2}}\right)\right]$$
where *μ*_1_
*μ*_2_ is the center of the IRFPA, Suppose $\sigma_{}$ equals to 50. The quantum conversion efficiency *η*(*λ*) and global gain *k_i,j_*(*λ*) are wavelength dependent functions. We assume that *η*(*λ*) and *k*(*λ*) of each pixel are approximately quadratic polynomial with one variable [18,19,20,21].
(15)$$G_{i,j}(\lambda {)=k}_{i,j}(\lambda )*\eta (\lambda )=a\lambda^{2}+b\lambda +c$$

We simulate the quadratic polynomial function of the response gain *G_i,j_* and wavelength on Equation (15), and simulate the response model at wavelength 3~15 μm, as shown in the Figure 3.

According to the Gaussian distribution characteristics of optical image intensity, it is assumed that the coefficients *a*, *b*, *c* of the quadratic polynomial satisfy normal distribution as follows.
(16)$$\{\begin{array}{c} a∼N\left(\mu_{1},\sigma_{1}^{2}\right),\mu_{1}=-0.025,\sigma_{1}=0.005\\ b∼N\left(\mu_{2},\sigma_{2}^{2}\right),\mu_{2}=0.45,\sigma_{2}=0.09\\ c∼N\left(\mu_{3},\sigma_{3}^{2}\right),\mu_{3}=-1.125,\sigma_{3}=0.225 \end{array}$$

The simulated global gain of the detector *k_i,j_*(*λ*), quantum efficiency *η*(*λ*), atmospheric transmittance *τ_s_*(*λ*), and radiation flux *Φ*(*λ*) received by each pixel were respectively substituted into the traditional model Equation (5) and the modified model Equation (13), and the deviations between the correction coefficients were calculated as shown in Figure 4.

The simulation results show that the correction coefficient of the proposed modified model is quite different from that of the traditional model after introducing the atmospheric transmittance to the spectral response factor of the infrared detector. The local correction coefficients of the traditional correction model have large deviations, which lead to the existence of fixed-pattern noise in the corrected image.

## 4. Experiment

To verify the effect of atmospheric transmittance on the NUC of the IRFPA, firstly, a response slope test of the infrared focal plane array was carried out in the laboratory. Secondly, we calculated the correction coefficients separately using the TPC method in two cases depending on whether or not an atmospheric transmittance equivalent analog filter was installed in the optical path, as shown in Figure 5. Finally, we observed the sky scene outdoors and used two sets of correction coefficients to correct the original images in the same scene for comparison experiments.

### 4.1. Experiment of TPC without Atmospheric Transmittance Filter

#### 4.1.1. Black Body as the Target Image

The most basic calibration method for a long-wave infrared detection system is the external surface source calibration method [22,23]. For this, the whole set of experimental equipment is placed on the optical vibration isolation platform, as shown in Figure 6, and the IRFPA parameters are shown in Table 1.

We adjusted the black body temperature from −20 to 10 °C, and collected a set of images at intervals of 5 °C. The corresponding relationship between the temperature and the average gray value of the images is shown in Figure 7.

Figure 7 and Table 2 show that the response linearity difference between average slope and linear fitted slope is 0.36%. The response shows good linearity in the temperature range of −20~10 °C. Taking the temperature points of −10 °C and +10 °C as the reference image, the original image output by the detector was collected, as shown in Figure 8.

Blind cells and FPN can be clearly seen in the original image. We next eliminated the blind cells and calculated the TPC coefficients, as shown in Figure 9 and Figure 10.

We then collected the original image of the black body at −20 °C as the target image, as shown in Figure 11a, and used the corresponding set of correction coefficients to correct this target image. The corrected image is uniform, as shown in Figure 11b.

The experimental results show that a uniform image is obtained when using the correction coefficients from the black-body reference image to correct the black-body target image.

#### 4.1.2. Sky Scene as Target Image

We aimed the LWIR detection system at the sky scene and corrected the target image using the correction coefficients provided in Section 4.1.1. The corrected image shown in Figure 12 presents serious fixed-pattern noise, even though the average grayscale value 6673DN is approximately equal to the average value of the −20 degree blackbody image.

Based on the theoretical analysis in Section 3, we can assume that this FPN phenomenon is caused by deviation from the correction coefficients directly obtained from the black body, which are not suitable for correcting sky images with remote atmospheric paths.

In order to verify this conjecture, we customized a filter to simulate the atmospheric transmittance of the actual experimental environment according to the response wavelength band of 8–12 μm and the simulation curve of scene 2 in Figure 3. The detailed parameters are shown in Table 3, and the transmittance curve of the filter is shown in Figure 13.

An atmospheric transmittance equivalent analog filter was placed between the IRFPA detector and rear lens, and the original images of the low-temperature blackbody at −10 °C and 10 °C were collected as reference images. We then calculate the TPC parameters of the system *K_i,j_* and *B_i,j_* with the filter, as shown in Figure 14 and Figure 15.

The overall trend of TPC coefficients of the two groups is consistent, but some details are quite different. The deviation is shown in Figure 16 and Figure 17.

To analyze the differences between the correction coefficients with and without the atmospheric transmittance equivalent analog filter more clearly, we selected the correction coefficients in the 190th column of the image for comparison, as shown in Figure 18.

Here, the correction coefficients with filters calculated for the same temperature blackbody have present roughly the same trend as those without filters, but there are obvious deviations in the details of some pixels. Therefore, the atmospheric transmittance characteristics will deviate from the correction coefficients.

### 4.2. Experiment of TPC in Sky Scene with Two Sets Correction Coefficients

In actual flight, we found that system imaging without the modified two-point correction method was very unsatisfactory with fixed-pattern noise clearly visible in Figure 19. In the scanning process for long-distance weak targets, there is a high probability of appearing in the position of fixed-pattern noise, which is extremely unfavorable to the detection of weak and small targets, seriously affecting the accuracy of detection.

To verify the conjecture that the influence of atmospheric transmittance on the deviation of the correction coefficients leads to the appearance of fixed-pattern noise, we compensated the optical path with an atmospheric transmittance equivalent analog filter. We obtained two sets of correction coefficients: with and without filters.

To compare the imaging quality of the sky background image with the two sets of correction coefficients, we placed the LWIR detection system on an open outdoor site to image the sky scene. Figure 20 shows the experimental scene.

We collected the original image of the sky scene and used two sets of correction coefficients to correct the target image via the TPC method. The original target image is shown in Figure 21.

The correction results for the same target image are shown in Figure 22a, and 3D plot of the image corrected using the correction coefficients without filter is shown in Figure 22c. It can be clearly seen that the corrected image obtained via the correction coefficients without the filter has obvious fixed-pattern noise. In the target image corrected using the correction coefficients with the filter in Figure 22b and 3D plot of corrected image of correction coefficients with filter in Figure 22d, can clearly see the gradient background in the sky. Additionally, the FPN is significantly reduced in the corrected image.

To evaluate the non-uniformity of an IRPFA quantitatively, a number of non-uniformity evaluation indicators have been proposed. At present, the most commonly used one is NU, which can be expressed as [24,25].
(17)$$NU=\frac{1}{\bar{Y}}\sqrt{\frac{1}{M\times N}\sum_{i=1}^{M}\sum_{j=1}^{N}(Y_{i,j}-\bar{Y}{)}^{2}}\times 100\%$$
where *Y_i,j_* is the gray value of pixel (*i, j*) in the corrected image, and $\bar{Y}$ denotes the average value of all pixels. Usually, the larger the value of *NU*, the worse the image non-uniformity.

When the system detects weak and small targets, the target imaging size generally does not exceed 5 × 5 pix. In practical application of airborne IRST, target movements tend to range from far to near. In order to analyze the effect of atmospheric transmittance on the non-uniform correction of the IRFPA more quantitatively, we calculated the 5 × 5 pix, 7 × 7 pix, and 9 × 9 pix local regional standard deviation (STD) for the corrected images of the two sets of correction coefficients pixel by pixel. The probability density curve and probability distribution curve of the local standard deviation are shown in Figure 23 and Figure 24.

The probability peak characteristics corresponding to the local STD and NU of different image sizes are listed in Table 4 [26].

As shown in Table 3, the local STD and NU values of the corrected image without filter are 2.5~3.9 times greater than the value with filter within a local area 9 × 9 pix. It can be seen that the atmospheric transmittance will affect the NUC performance of the infrared image. When correcting the IRFPA via the two-point correction method, the local image STD can be effectively reduced by fully considering the atmospheric transmittance characteristics of the system working scene.

## 5. Discussion

Using the modified two-point correction (TPC) method, we carried out a flight experiment to detect small and weak targets in the air. The heights of the target aircraft and carrier aircraft were 5 km, and the distance was 30 km. The image of the target aircraft is detected in the Figure 25, and the target can be clearly seen in the image. Through the comparison of two flight tests, we determined that even under a complex cloud background, the image corrected using the correction coefficients without filter still had strong FPN, while in the image corrected using the correction coefficients with filter, FPN is barely visible.

Although the atmospheric transmittance equivalent analog filter and the actual working atmospheric environment will have certain differences and cannot eliminate the fixed-pattern noise (FPN) completely, it can still be qualitatively seen that the FPN is effectively weakened.

In the future work, we will simulate the atmospheric transmittance curve according to the actual environment and develop more accurate equivalent analog filters. We also plan to combine the modified TPC method with the weekly sweep operation mode of airborne photoelectric detection equipment to further improve the quality of NUC and further reduce the FPN.

## 6. Conclusions

Through the above theoretical analysis and experimental comparison, we determined that atmospheric transmittance has a crucial influence on the two-point correction in the LWIR detection system. When design an LWIR detection system, it is necessary to simulate the atmospheric transmittance design according to the working scene of the system. By designing an equivalent analog device for simulating atmospheric transmittance, the system becomes closer to the real working scene when performing two-point calibration on the LWIR detection system, and the fixed-pattern noise of the IRFPA is more effectively reduced. Ultimately, full consideration of the influence of atmospheric transmittance on infrared NUC can effectively improve the signal-to-noise ratio of weak and small target detection in LWIR detection systems.

## Figures and Tables

**Figure 1 sensors-23-02054-f001:**
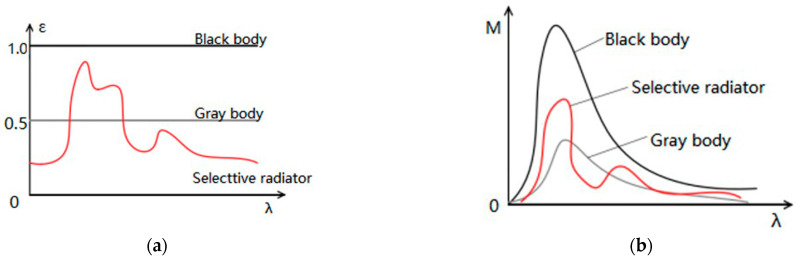
Distribution curves of spectral emissivity and spectral radiant output of black body, gray body and selective radiator: (**a**) Comparison of emissivity; (**b**) Comparison of radiation exitance.

**Figure 2 sensors-23-02054-f002:**
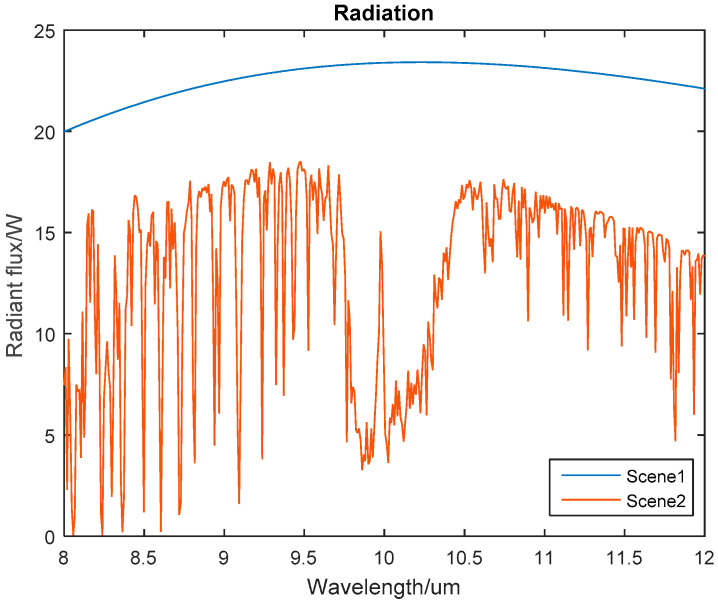
The effect of atmospheric transmittance on radiation: Working scene 1 is the black body calibration of the laboratory environment; working scene 2 is the sky scene where the ground observation sky height is 50 km and the target distance is 55 km.

**Figure 3 sensors-23-02054-f003:**
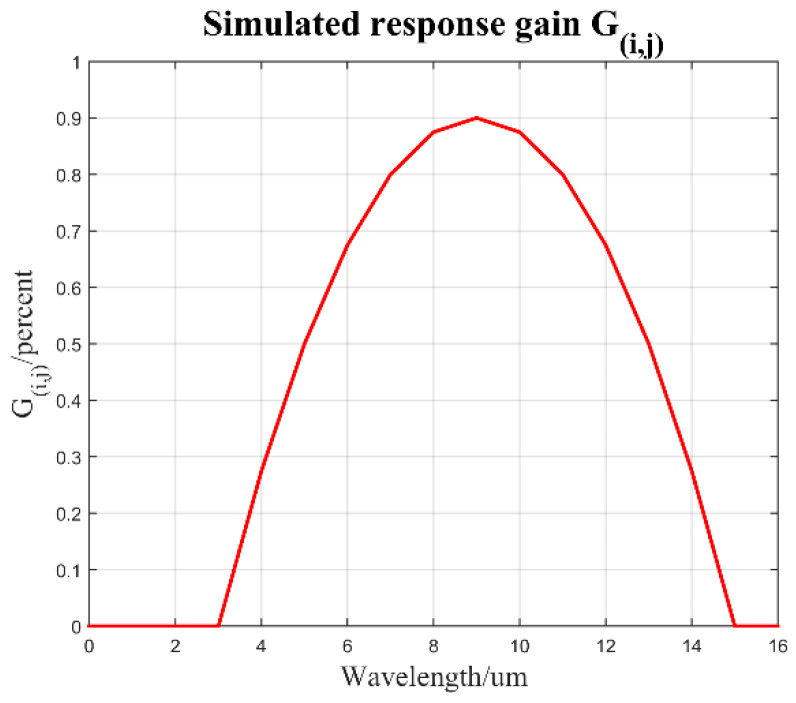
The simulated response gain *G_i,j_*.

**Figure 4 sensors-23-02054-f004:**
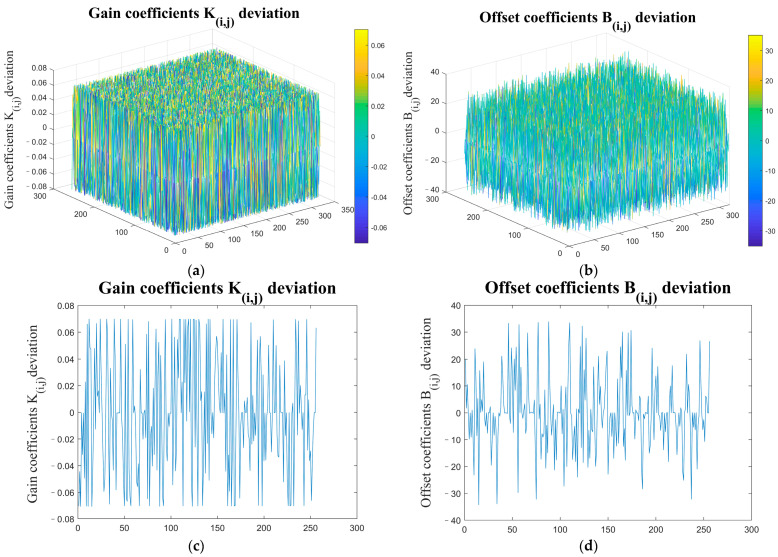
Comparison of correction coefficients between traditional model and modified model: (**a**) correction gain coefficients *K_i,j_* deviation; (**b**) correction offset coefficients *B_i,j_* deviation; (**c**) correction gain coefficients *K_i,j_* deviation in 190th column; (**d**) correction offset coefficients *B_i,j_* deviation in 190th column.

**Figure 5 sensors-23-02054-f005:**
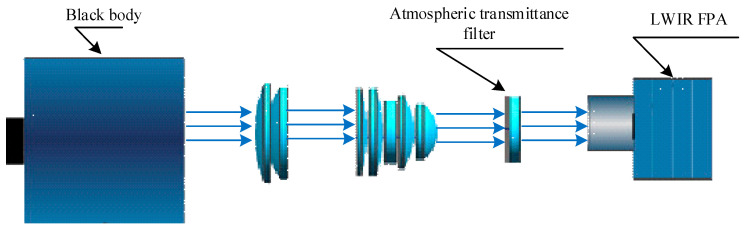
Optical path diagram of the experimental system.

**Figure 6 sensors-23-02054-f006:**
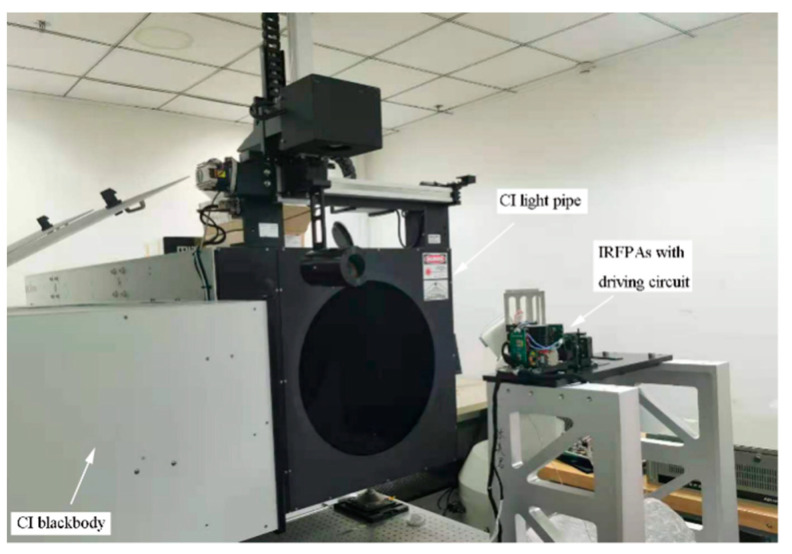
Experiment for TPC: Testing the response slope of the IRFPA And calculating the correction coefficients with and without the atmospheric transmittance filter.

**Figure 7 sensors-23-02054-f007:**
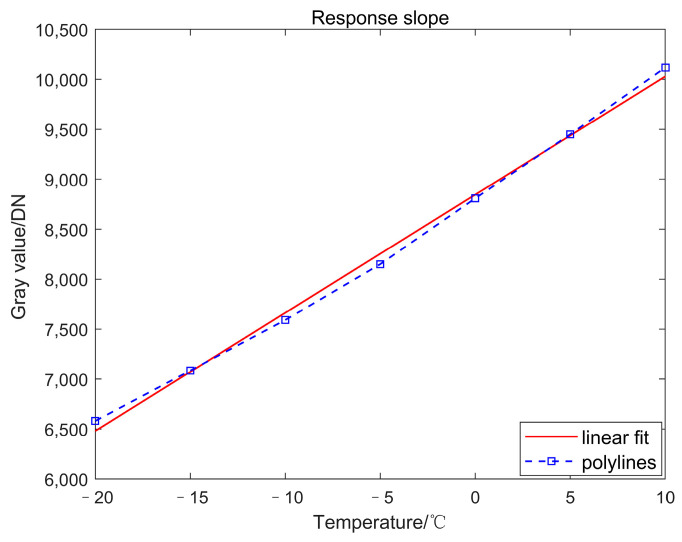
Response slope of IRFPA.

**Figure 8 sensors-23-02054-f008:**
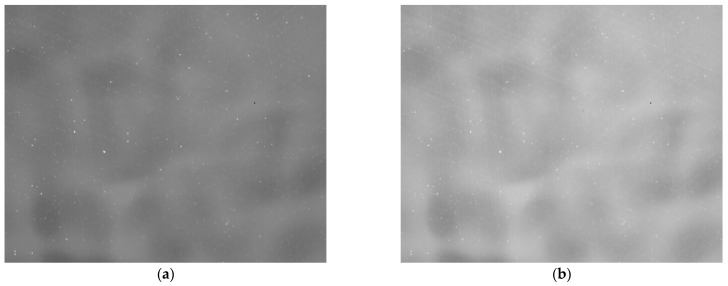
Blackbody reference raw images for low and high temperatures. (**a**) −10 °C reference image (average gray value 7591DN); (**b**) 10 °C reference image (average gray value 10117DN).

**Figure 9 sensors-23-02054-f009:**
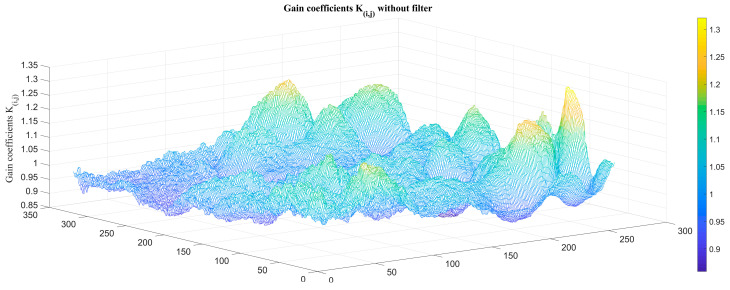
TPC gain coefficients *K_i,j_* without atmospheric transmittance filter.

**Figure 10 sensors-23-02054-f010:**
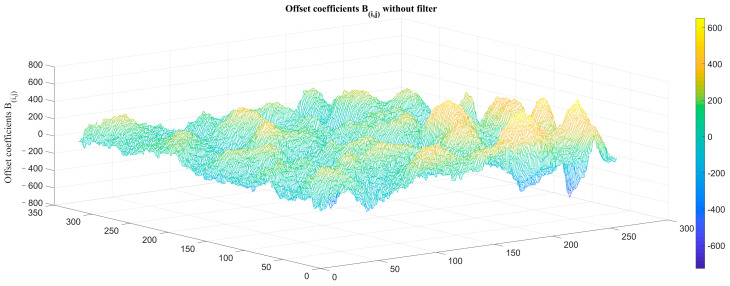
TPC offset coefficients *B_i,j_* without atmospheric transmittance filter.

**Figure 11 sensors-23-02054-f011:**
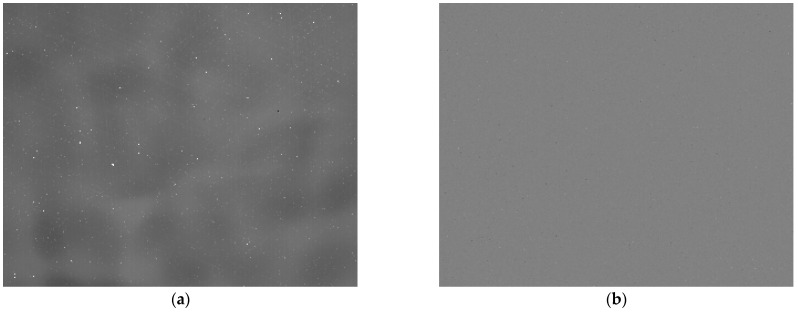
Target image and corrected image: (**a**) target image at −20 °C (black body average gray value 6581DN); (**b**) corrected image without atmospheric transmittance.

**Figure 12 sensors-23-02054-f012:**
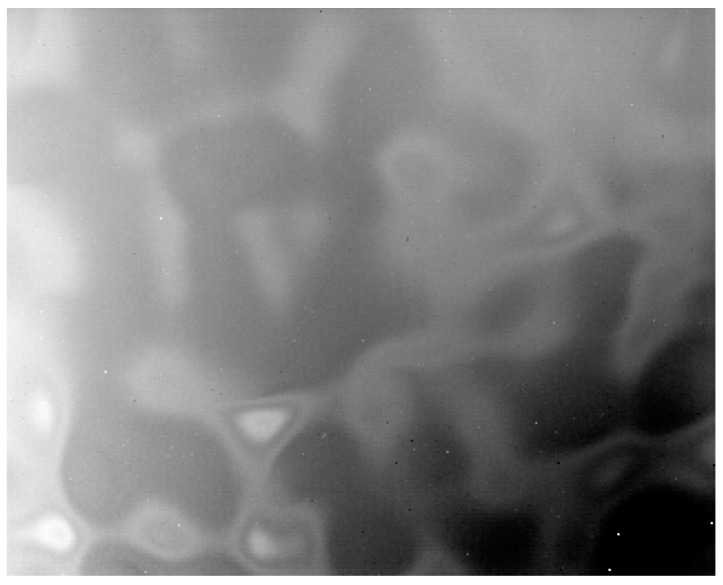
Corrected sky image with fixed-pattern noise. (average grayscale value of 6673DN, which is close to the −20 °C black body temperature corrected using the coefficients in Figure 7 and Figure 8).

**Figure 13 sensors-23-02054-f013:**
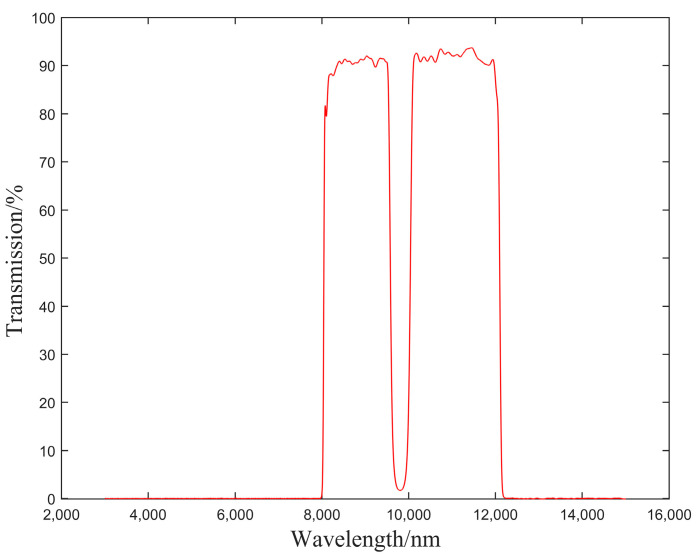
Transmittance curve of atmospheric transmittance equivalent analog filter.

**Figure 14 sensors-23-02054-f014:**
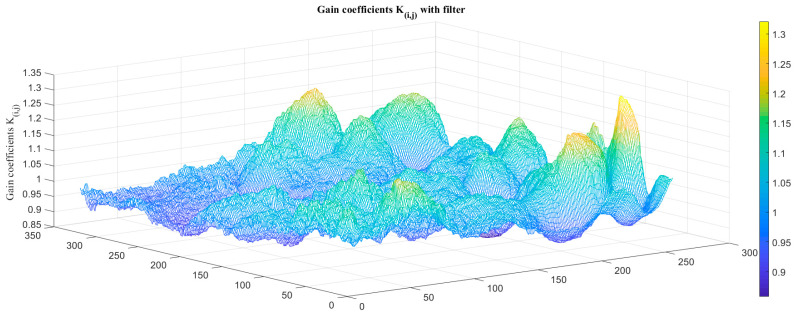
TPC gain coefficients *K_i,j_* with atmospheric transmittance filter.

**Figure 15 sensors-23-02054-f015:**
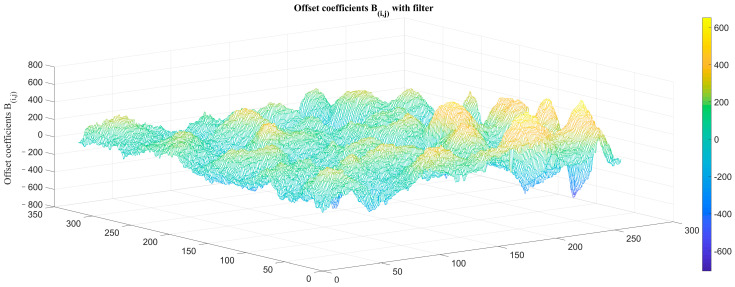
TPC offset coefficients *B_i,j_* with atmospheric transmittance filter.

**Figure 16 sensors-23-02054-f016:**
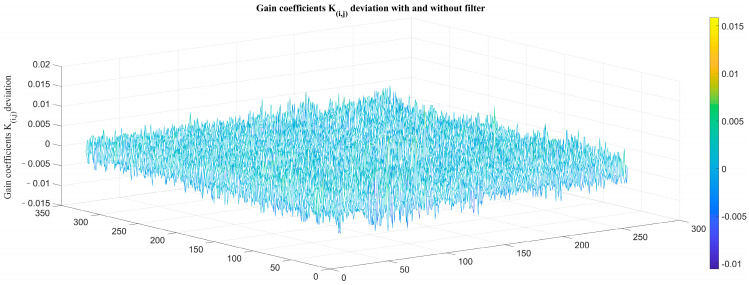
Deviation of TPC gain coefficients *K_i,j_* with and without atmospheric transmittance equivalent analog filter.

**Figure 17 sensors-23-02054-f017:**
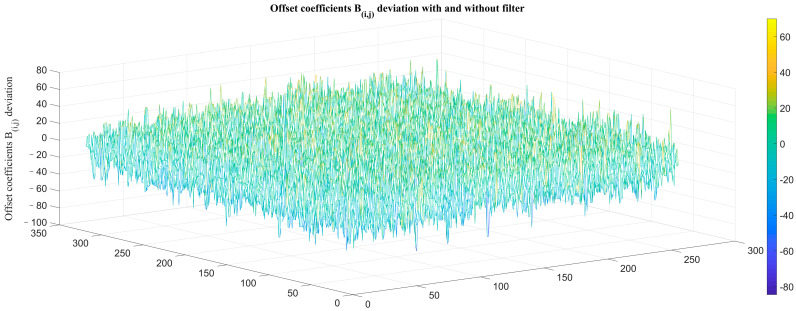
TPC offset coefficients *B_i,j_* deviation with and without atmospheric transmittance equivalent analog filter.

**Figure 18 sensors-23-02054-f018:**
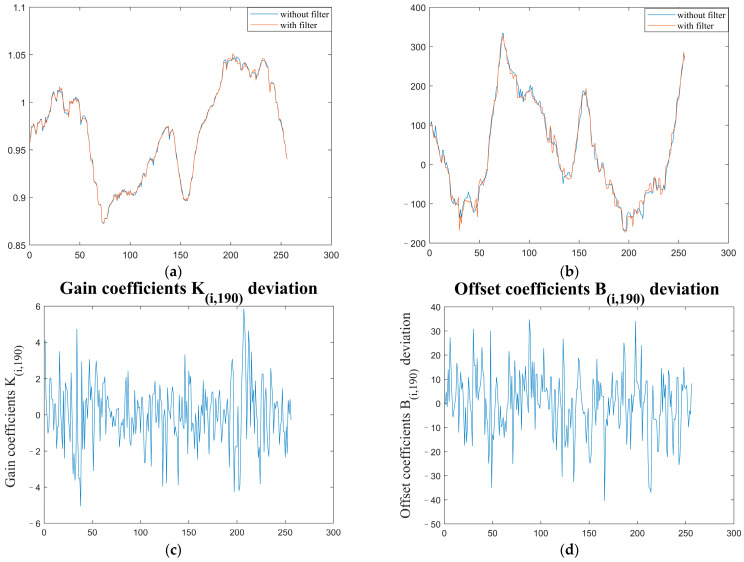
Comparison of correction coefficients in the 190th column: (**a**) correction gain coefficients *K_i,j_* in the 190th column; (**b**) correction offset coefficients *B_i,j_* in the 190th column; (**c**) correction gain coefficients *K_i,j_* deviation in 190th column; (**d**) correction offset coefficients *B_i,j_* deviation in 190th column.

**Figure 19 sensors-23-02054-f019:**
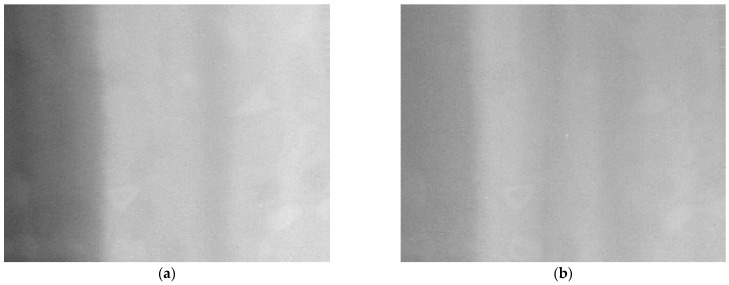
Flight experiment of detecting small and weak target without filter. (**a**) clouds and skyline; (**b**) close-range non-cooperative goals.

**Figure 20 sensors-23-02054-f020:**
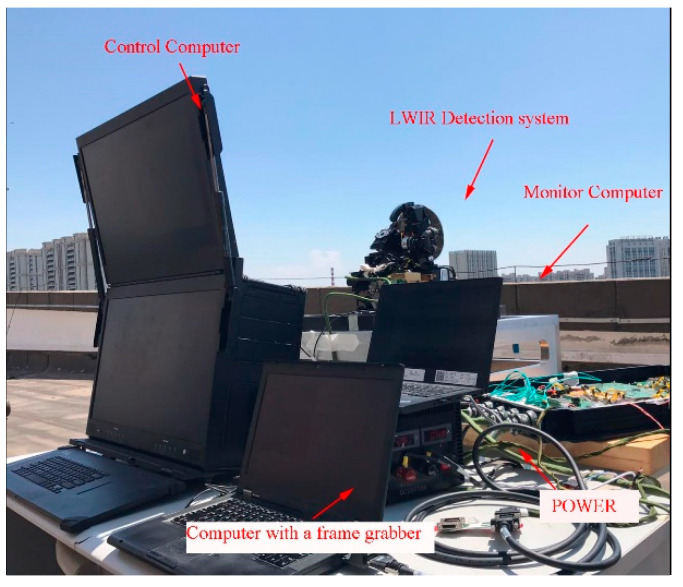
Experimental sky scene with two sets of correction coefficients.

**Figure 21 sensors-23-02054-f021:**
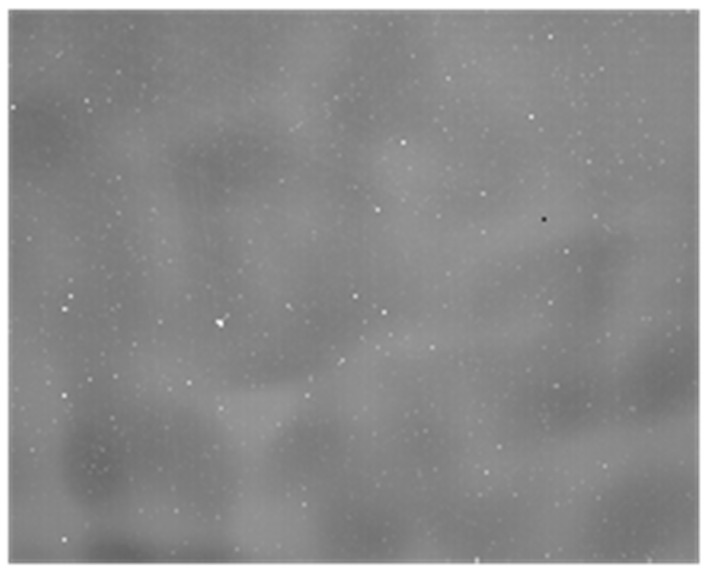
Target image of the sky scene to be corrected by the two sets of correction coefficients.

**Figure 22 sensors-23-02054-f022:**
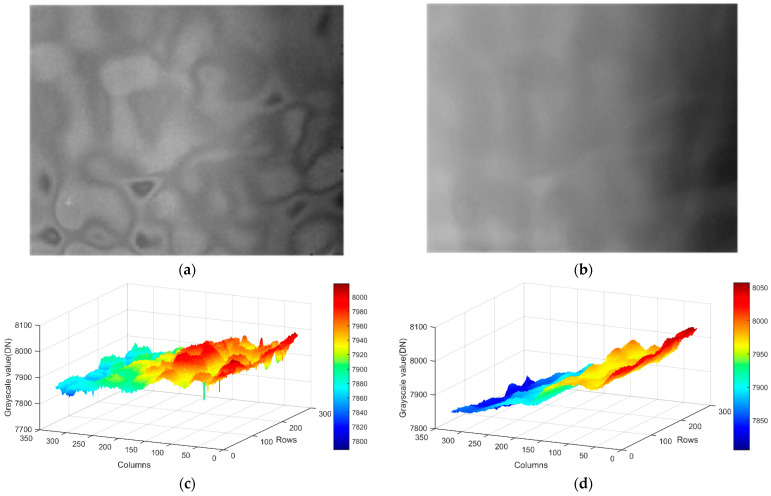
Comparison of same target image corrected by two sets of correction coefficients: (**a**) image corrected using the correction coefficients without filter; (**b**) image corrected using the correction coefficients with filter; (**c**) 3D plot of the image corrected using the correction coefficients without filter; (**d**) 3D plot of the image corrected using the correction coefficients with filter.

**Figure 23 sensors-23-02054-f023:**
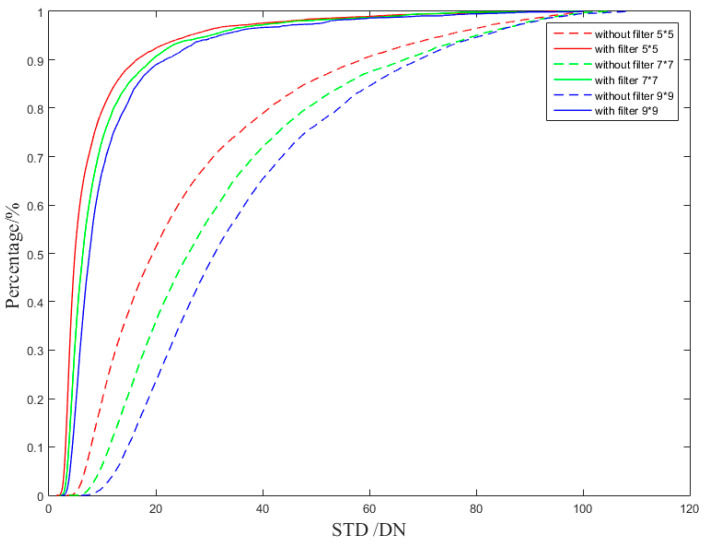
Local regional standard deviation (STD) probability density curve comparison of the corrected images.

**Figure 24 sensors-23-02054-f024:**
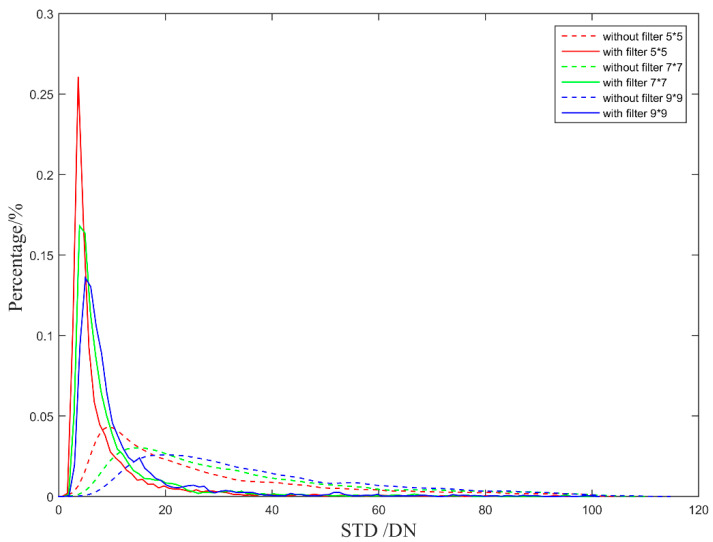
Local regional standard deviation (STD) probability distribution curve comparison of the corrected images.

**Figure 25 sensors-23-02054-f025:**
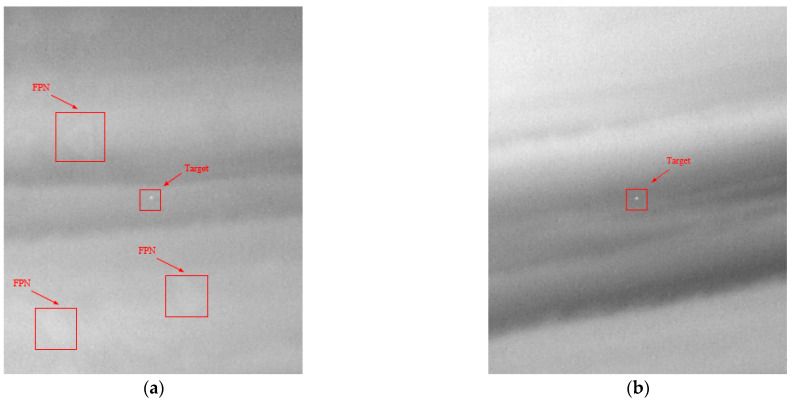
Flight experiment detect small and weak targets with application of image correction coefficients: (**a**) without filter; (**b**) with filter.

**Table 1 sensors-23-02054-t001:** IRFPA parameters.

Parameters	Value
Spectral band	7.7–11.3 μm
Material	HgCdTe
Resolution	320 × 256
NETD	19 mK
Bit depth	14 bit
Focal length	38 mm
F/#	2

* F/# is the F-number

**Table 2 sensors-23-02054-t002:** Response linearity of IRFPA for the range −20~10 ℃.

Parameters	Value
Average slope	117.87
Linear fitted slope	118.30
Linearity difference	0.36%

**Table 3 sensors-23-02054-t003:** Atmospheric transmittance equivalent analog filter coefficients.

Description	Requirement
Transmission	T > 85 ± 1% (for the range 8.0–9.5 µm)
Transmission	T > 85 ± 1% (for the range 9.9–11.7 µm)
Reflection Average	R > 94 ± 1% T < 5%(for the range 9.6–9.8 µm)
Out-of-Band Blocking	T < 0.1% average 3–15 µm
Size 1	Diameter1 = 25 mm
Clear Aperture 1	C1 ≥ 20 mm
Size 2	Diameter2 = 16 mm
Clear Aperture 2	C2 ≥ 13 mm
Thickness	1 mm nominal
Surface Quality	E-E per Mil-C-48497A (60/40 equivalent)
Pinholes	Best practices to minimize pinholes, no guarantee
Construction	Unmounted, single substrate
TWF Error	TWF < 1/4 wave P-V per inch—prior to coating customer accepted 2 waves P-V
Parallelism	Parallelism < 20 arc seconds
Substrate material	Optical grade germanium

**Table 4 sensors-23-02054-t004:** Comparison of peak characteristics for local STD peaks and NU of different image sizes.

	Corrected Image without Filter	Corrected Image with Filter
Local STD (5 × 5)	9.19	3.64
Local STD (7 × 7)	13.91	3.89
Local STD (9 × 9)	19.57	4.98
Average grayscale	7922.2	7922.1
Local NU (5 × 5)	0.116%	0.0459%
Local NU (7 × 7)	0.176%	0.0491%
Local NU (9 × 9)	0.247%	0.0629%

## Data Availability

Not applicable.

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
