# Peer review of "Modified Two-Point Correction Method for Wide-Spectrum LWIR Detection System"

_sensors, 2023, doi:10.3390/s23042054_

Round 1

Reviewer 1 Report

The authors propose the use of an atmospheric transmittance filter to improve the performance of conventional black body based LWIR sensor calibration using two-point correction. They show the performance enhancement through several experiments.

However, I think the manuscript is written as like technical report rather than journal paper. This manuscript should be much improved for possible publication, in terms of technical and English writing.

Some minor corrections and concerns are as follows;

All references have to be cited in appropriate places.

42nd line on page 1: changes “consist of” to "are”.

47th line on page 2: What stands for IRST?

79th line on page 2: changes “frequency band” to “wavelength”.

121st line on page 3: changes “np” to “n_p”.  

125th line on page 3: changes “ne” to “n_e”.  

163rd line on page 5: What is “k_i,j” in the denominator in the equation for K_i,j ?

290~293rd lines on page 12~13: It would be better to show other sky scene images not using filter, for the authors to insist a fixed pattern noise.

298th line on page 13: change “without” to “with” in the caption for figure (b).

The deviation of gain and offset parameters with/without filter looks like big in Fig. 12 and 13, but it is very small in Fig. 14. It is required to explain the inconsistence. 

Author Response

Response to Reviewer 1 Comments

Thank you very much for taking your time to review this manuscript. I appreciate all your comments and suggestions. Please find my itemized responses below and my revisions in the re-submitted files.

Comments and Suggestions for Authors

Point 1: The authors propose the use of an atmospheric transmittance filter to improve the performance of conventional black body based LWIR sensor calibration using two-point correction. They show the performance enhancement through several experiments.

However, I think the manuscript is written as like technical report rather than journal paper. This manuscript should be much improved for possible publication, in terms of technical and English writing.

Response 1: Our study aims at the advancement of the related scientific field. With regard to this, we mainly improved the paper mainly improved the manuscript according to the following three aspects:

  • The need for more in-depth investigation of the two-point correction method is more clearly outlined in the introduction.
  • The problems and research methods are explained in more detail.
  • Multi-scale non-homogeneity analysis is supplemented by experimental analysis. In the practical application of airborne IRST, high-speed moving targets tend to gradually change in size, so the characteristics of 7×7pix, 9×9pix local regional STDs are supplemented in Figure 21 and 22, Table 4. The modified two-point correction method in this paper is more applicable to the improvement of the signal-to-noise ratio in the local regional.
  • Extensive editing of English language was improved by language editing services.

Some minor corrections and concerns are as follows;

Point 2: All references have to be cited in appropriate places.

Response 2: Along with the guidelines given by Sensors, I reorganized the format of the references and added many DOI-links as possible.

Point 3: 42nd line on page 1: changes “consist of” to "are”.

Response 3: Based on your guidance and suggestions, the error has been corrected.

Point 4: 47th line on page 2: What stands for IRST?

Response 4: IRST means infrared search and track (IRST) systems. This term is mentioned in line 30th on the first page.

Point 5: 79th line on page 2: changes “frequency band” to “wavelength”.

Response 5: Based on your guidance and suggestions, the error has been corrected.

Point 6: 121st line on page 3: changes “np” to “n_p”.  125th line on page 3: changes “ne” to “n_e”.  

Response 6: Based on your guidance and suggestions, the errors have been corrected. Changed “np” to “np”, changed “ne” to “ne”.

Point 7: 163rd line on page 5: What is “k_i,j” in the denominator in the equation for K_i,j ?

Response 7: I am so sorry for my mistake,  , where k is the global linear gain of the detector, and k_i,j is a function of λ, The first stage of derivation of the formula is to assume that the full cell response is wavelength-independent , so k_i,j is not a function of λ function outside the integral. In fact, the spectral response rate of each cell is different, k_i,j is a function of λ, so in the second stage of formula derivation it must to be changed to k_i,j(λ),and put into the integral.

Thank you very much for correcting this error in my most important formula. Your professionalism and academic knowledge are very admirable.

Point 8: 290~293rd lines on page 12~13: It would be better to show other sky scene images not using filter, for the authors to insist a fixed pattern noise.

Response 8: Two images of experimental flight tests are cited in 317th Figure 17. This images contain clouds, skyline, and close-range non-cooperative goals, and clearly show fixed-pattern noise.

Point 9: 298th line on page 13: change “without” to “with” in the caption for figure (b).

Response 9: Thanks for your suggestion. This was my mistake, and “without” has been changed to “with” in 343rd line Figure 20.

Point 10: The deviation of gain and offset parameters with/without filter looks like big in Fig. 12 and 13(new Figures 14&15), but it is very small in Fig. 14(new Figure 16). It is required to explain the inconsistence.

Response 10: The 190th column of the image is offered to allow comparison and we subtract the two sets of data, so that we can see the difference between the two sets of data much more clearly in Figure 16.

Reviewer 2 Report

The manuscript presents a theoretical analysis and experimental study on the effect of atmospheric transmittance characteristics of long-range paths on the non-uniformity of wide-spectrum long-wave infrared systems has not been studied. Also, it introduces a modified two-point correction method by designing an equivalent analog filter for simulating atmospheric transmittance. I have the following comments for the authors:

·         The manuscript needs English improvements. I suggest avoiding long sentences.

·         Figures 7, 8, and 11 need a y-axis label.

·         In Figure 17-b caption, I believe the authors mean “with filter” not “without”.

·         In Line 315, the reference to Table 3 is missing.

·         In Line 342, I suggest replacing “In the following study” with “In future work”.

Author Response

Response to Reviewer 2 Comments

Thank you very much for taking your time to review this manuscript. I appreciate all your comments and suggestions. Please find my itemized responses below and my revisions in the re-submitted files.

Comments and Suggestions for Authors

The manuscript presents a theoretical analysis and experimental study on the effect of atmospheric transmittance characteristics of long-range paths on the non-uniformity of wide-spectrum long-wave infrared systems has not been studied. Also, it introduces a modified two-point correction method by designing an equivalent analog filter for simulating atmospheric transmittance. I have the following comments for the authors:

Point 1: The manuscript needs English improvements. I suggest avoiding long sentences.

Response 1: Thank you for your suggestion, I have attempted to improve the English through editing.

Point 2: Figures 7, 8, and 11 need a y-axis label.

Response 2: Thank you for your suggestion. In the new version of these figures, y-axis label was added.

Point 3: In Figure 17-b (new Figure 20) caption, I believe the authors mean “with filter” not “without”.

Response 3: Yes, thank you for your suggestion. This is my mistake, and “without” has been changed to “with” in Figure 20.

Point 4: In Line 315, the reference to Table 3(new Table 4) is missing.

Response 4: Thank you for your suggestion. I refer to the method of calculating STD in this article “Scene-Based Nonuniformity Correction for Airborne Point Target Detection Systems”, and the author used to be in the same lab with me. The reference has now been cited.

Point 5: In Line 342, I suggest replacing “In the following study” with “In future work”.

Response 5: “In the following study” has been changed to “In future work”.

Round 2

Reviewer 1 Report

The manuscript is now modified in parts according to the comments mentioned in the 1st round of review. However, the manuscript still seems like a technical report rather than a journal paper. I recommand that an academical analysis is added in the final version.

Author Response

Response to Reviewer 1 Comments

Thank you very much for taking your time to review this manuscript. I appreciate your comments and suggestions. Please find my itemized responses below and my revisions in the re-submitted files.

Comments and Suggestions for Authors

Point 1: The manuscript is now modified in parts according to the comments mentioned in the 1st round of review. However, the manuscript still seems like a technical report rather than a journal paper. I recommend that an academical analysis is added in the final version..

Response 1: In Section 3, we add a subsection 3.2 Performance analysis, which include the simulation analysis between the modified model and the traditional model, and theoretically explained that the modified model is the correction of the traditional model by introducing the atmospheric transmittance to the spectral response factor of the infrared detector.

In order to show the correction of the modified model to the traditional model more scientifically, we refer to the existing detector parameters and relevant literature, add quantitative simulation analysis for the correction coefficients of the two groups of models, and simulate the deviation between the two groups of correction coefficients. The simulation results prove that there are deviations between the correction coefficients of the traditional model and the improved model.

The simulation results show a similar trend to the experimental results, which illustrates that the local correction coefficients of the traditional correction model have large deviations, which lead to the existence of fixed-pattern noise in the corrected image.